# Quality of Life Among Latino/a Adults: Examining the Serial Mediation of Network Acculturation, Psychological Acculturation, Social Capital, and Helping-Seeking

**DOI:** 10.3390/bs15030388

**Published:** 2025-03-19

**Authors:** Adrian J. Archuleta, Stephanie Grace Prost, Mona A. Dajani

**Affiliations:** Kent School of Social Work and Family Science, University of Louisville, Louisville, KY 40292, USA; stephanie.prost@louisville.edu (S.G.P.); mona.dajani@louisville.edu (M.A.D.)

**Keywords:** acculturation, help-seeking, Latinos/as, social capital, quality of life, networks

## Abstract

Latinos/as are the largest ethnic group in the U.S. and are a continuous source of population growth. Therefore, their health and quality of life are important public health concerns. Acculturation is an important determinant of health for Latinos/as. However, few studies examine models identifying determinants of acculturation along with its relationship to other social and health behaviors. The current study uses social network data from a sample of crowdsourced recruited Latinos/as (*N* = 300) to examine a structural model between network acculturation, psychological acculturation, social capital, help-seeking, and quality of life (QoL). The model posits several paths through which social networks (i.e., network acculturation) relate to acculturation and other model variables. Directly, network acculturation was found to be significantly related to Latino/a enculturation (−0.83, *p* = 0.002) and White American Acculturation (0.47, *p* = 0.003). Latino/a enculturation was related to help-seeking (0.21, *p* = 0.029) and social capital (0.36, *p* < 0.001), while White American acculturation was only related to social capital (0.35, *p* = 0.003). Social capital demonstrated a robust relationship with help-seeking (0.48, *p* = 0.004) and QoL (0.96, *p* = 0.003). The findings suggest that determinants of acculturation (i.e., network acculturation) are meaningful contributors to psychological acculturation and other variables relating to Latino/as’ QoL.

## 1. Introduction

Latinos/as represent 19% of the U.S. population and experience tremendous longevity, with life expectancy well into their 80s ([88]). Despite this longevity, Latinos/as are likely to work in jobs with increased health risks, are less likely to receive quality medical care, and are more likely to have preexisting health conditions, experience discrimination that hinders medical casre access, and live in communities with environmental concerns ([38]). Increases in longevity and the potential for poorer health elevate the importance of understanding Latinos/as’ quality of life (QoL), a crucial subjective measure related to one’s health beyond disease ([96]). The [98] ([98]) defines QoL as “an individual’s perception of their position in life in the context of the culture and value systems in which they live and in relation to their goals, expectations, standards and concerns (p. 11)”. A person’s perception of their position in life and their associated satisfaction is driven by complex factors that vary from group to group and the context in which they live and work.

Social relationships are an important facet of a person’s quality of life, with more extensive networks providing greater access to needed resources. Social capital is critical to a person’s health and quality of life as it mitigates stressful experiences, provides known benefits that improve health, and facilitates social mobility ([6]; [26]; [66]). Social capital is defined as the aggregated resources obtained through one’s membership in a group ([20]) and is often distinguished based on network, cognitive, and social integration descriptions, with each articulating how different facets of social capital are developed and maintained through one’s relationships ([66]). Social integration provides critical access to economic, political, cultural, and social resources that are associated with health determinants that drive inequities between groups ([73]).

Acculturation has been identified as a factor associated with Latinos/as’ social integration ([36]) and access to social capital ([77]; [19]; [90]). Acculturation is defined as the cultural changes that occur when two distinct cultural groups interact for a sustained period, resulting in changes in one or both groups ([75]). Acculturation has been described as degrees of change associated with strategies (e.g., assimilation, integration, separation, and marginalization) enacted by individuals and their communities that create specific acculturative experiences ([18]). Changes in acculturation can alter a person’s social networks, including family relationships ([50]), which have meaningful implications for the type of social capital available. For example, different aspects of acculturation (e.g., social versus linguistic) and associated experiences (e.g., acculturative stress) shape network size, frequency of contact, trust, and social influence ([10]). Therefore, differences in social capital characteristics associated with acculturation have important implications for their health and quality of life.

Help-seeking is also important in maintaining one’s physical and mental health. An unwillingness to access help while experiencing psychological distress could exacerbate existing symptoms or result in the loss of life (e.g., unaddressed suicidal ideations). Help-seeking is influenced by various individual, social, and cultural factors ([97]); however, these factors are under-explored among Latinos/as. Acculturation and social capital (e.g., perceived trust) play critical roles in the help-seeking process, with each often determining the pathways through which care is received. For example, differences in help-seeking are often observed between first- and later-generation ethnocultural groups, with less acculturative groups preferring informal sources (e.g., friends) for assistance ([32]). Collectively, these determinants of health are important in addressing the health needs of Latino/a adults and likely enhance aspects of a person’s life that increase quality and satisfaction.

### 1.1. Quality of Life (QoL)

While medical interventions have increased human longevity, they have not eliminated diseases and chronic illnesses that erode the quality of one’s life ([43]; [69]; [74]). Increasingly, QoL is seen as an important outcome of treatment ([101]) that helps assess the effectiveness of health services and associated policies ([15]; [30]). A person’s QoL encompasses aspects of physical and psychological health, social relationships, environment, personal beliefs, and independence ([96]). Quality of life has been examined in relation to a broad range of health conditions, including cancer, mental illness, heart disease, kidney disease, multiple sclerosis, diabetes, fibromyalgia, eye disease, pain, and obesity ([45]). Physical and mental health problems often restrict a person’s ability to function readily, influencing their overall satisfaction with life.

Despite the applicability of quality of life to all age groups, a substantial portion of the literature has focused on the experiences of older adults. However, the compounding effects associated with physical and mental health difficulties earlier in life warrant exploration. For example, among a crowdsourced sample in the United States, [11] ([11]) found that young and middle adults reported lower QoL scores than reported in the literature for other populations. Acculturation has demonstrated mixed results in predicting QoL among Latinos/as, with assimilation seeming to produce the best outcome ([86]). However, acculturation likely exerts an indirect effect on QoL through other meaningful factors that dictate access to important resources (e.g., income, social integration, health, and mental health literacy), suggesting that more sophisticated models are needed to understand these relationships.

### 1.2. Acculturation

Cross-cultural psychological theories have been relatively dominant in describing acculturation experiences. Psychological acculturation emphasizes the internal cultural changes that result from intercultural interaction ([39]). [102] ([102]) contends that acculturative changes are experienced at different depths, consisting of superficial (e.g., food, dress, and media use), intermediate (e.g., language use and preference for friends and neighbors), and significant (e.g., values, norms, and identity) levels. According to [16] ([16]), individuals develop strategies to preserve a desired amount of their heritage culture while engaging in a particular level of interaction with members of the cultural majority. High and low retention levels and interaction topologically form four strategies (e.g., assimilation, integration, separation, or marginalization) that reflect a person’s social integration and cultural change ([18]). For example, with assimilation, individuals desire to engage in high levels of interaction with individuals from the cultural majority while minimizing one’s heritage ([17]). Conversely, individuals who desire to maintain their cultural heritage while minimizing interaction with the cultural majority use separation as their acculturation strategy ([17]), reflecting their initial socialization to their heritage culture (i.e., enculturation; [93]). Integration occurs when individuals seek a balance or high levels of cultural retention and interaction with individuals from the cultural majority ([16]). Lastly, [17] ([17]) suggests that individuals who minimize the retention of their cultural heritage but are rebuffed by individuals from the cultural majority are marginalized.

For many groups, integration has been identified as the most adaptive acculturation strategy ([27]), but not for Latinos/as. Historically, separation has often been the most adaptative strategy because of the Latino/a diaspora (i.e., resettlement in traditional locations with dense Latino/a populations) in the U.S. and protective factors (e.g., social support) associated with Latino/a culture ([36]). More recently, Latinos/as have resettled in U.S. communities with previously small or absent Latino/a populations ([54]), which is associated with a greater need for adaptation and integration ([12]). Therefore, other acculturation strategies may be more viable as means of adapting than before, requiring continued investigation.

Due to its ubiquitous influence, acculturation has been identified as an essential cultural and social determinant because of its role in altering a person’s behaviors and health ([3]). Acculturative stress is often the driving force behind physical and mental health problems, which emerge from difficulties in reconciling cultural changes or conflict occurring within relationships ([18]). Social relationships are critical in facilitating acculturation ([16]; [18]; [59]; [92] and may be an important determinant of positive and negative acculturative experiences). However, scholars studying acculturation have called for more radical approaches that take relationships further and consider how the social forces associated with a person’s social networks contribute to changes in acculturation ([92]).

Incorporation into social networks is seen as a potential marker of structural assimilation, or “large-scale entrance into cliques, clubs, and institutions of host society, on a primary group level” ([42]). Gordon’s theory of assimilation describes the behavioral and attitudinal changes experienced but includes other potential measures that suggest broader change occurring in social life (e.g., structural assimilation and marital integration). For example, using Gordon’s definition of structural assimilation, [94] ([94]) examined a person’s neighborhood, organization membership and participation, and friendships. While Gordon’s theory has been criticized for its unidimensionality and inability to explain the experiences of non-European immigrants, modernizing (e.g., exploring social networks) several components may prove useful in explaining facets of acculturation not sufficiently explored in existing acculturation theory.

Social networks and their associated dynamics are important drivers of acculturation processes ([60]). Network acculturation refers to the interconnected social relationships that form a part of a person’s social network and contribute to their acculturation experiences ([89]). The maintenance of these relationships requires individual and collective balance to achieve cultural fit ([41]). Network approaches to acculturation draw on social network and social convoy theories, which describe how individuals form relationships that serve as context for a person’s acculturative experiences ([12]). Language use across stressful relationships has been used to measure network acculturation, which demonstrates a negative association with psychological measures of Latinx enculturation (i.e., socialization to one’s heritage culture) and a positive association with White American acculturation, suggesting that network acculturation may serve as a potential determinant for psychological experiences ([7]; [12]), highlighting the importance of social relationships in determining acculturation experiences.

### 1.3. Social Capital

Who people are connected to and how they are connected has been increasingly important in health research, as it determines the forms of stress a person is exposed to (e.g., life events and chronic stress) as well as the resources that are available to manage these and other life experiences ([66]). Social capital is described as a multidimensional construct characterized by trust, a sense of reciprocity, and levels of interconnectedness and integration that aid in mobilizing needed resources ([49]; [84]). Different types of networks (e.g., bonding, bridging, and linking) facilitate access to distinct resources ([55]). Bonding networks often reflect personal ties involving friends, family, and intimate partners, while bridging networks involve weaker ties with coworkers, neighbors, or community members ([58]). Bonding ties are more likely to provide access to emotional support, while bridging ties may provide access to opportunities beyond one’s immediate network.

Social relationships are an important bridge between acculturation, social capital, and social networks ([4]; [29]; [50]; [76]; [90]). Acculturation has been identified as a factor that changes a person’s relationships and social networks ([50]). [90] ([90]) found that higher levels of acculturation were associated with greater social capital and reported access to needed services. [71] ([71]) reported a more conditional relationship with acculturation moderated by social cohesion (i.e., a proxy for social capital) and negative social interaction, with higher acculturation and social cohesion associated with better mental health, and higher negative interaction and acculturation associated with worse mental health. In the context of acculturation, integration into both cultural groups may provide greater access to resources that maximize a person’s social capital. Identifying the role that acculturation plays in altering social capital is critical to understanding how acculturative experiences contribute directly and indirectly to a person’s quality of life.

Social capital is recognized as a key determinant of QoL ([68]), suggesting that a person’s social environment offers benefits across areas of life that are important to perceived quality and satisfaction ([23]). Despite its salutogenic effects, social capital has demonstrated an inconsistent relationship with QoL, often relating to a person’s mental health, but not their physical health ([40]). One overlooked area of research involves the connection between a person’s culture and social capital. Experiences with acculturation have been observed to alter a person’s social networks, which affects access to social capital. For example, [50] ([50]) reported that the familial network size and frequency of interaction of Mexican Americans generationally decreased, suggesting that increased exposure to American culture changes the qualities of a person’s social networks. Changes in social networks are important as they may alter the pathways typically used during mental health crises to identify critical forms of help.

### 1.4. Help-Seeking

Help-seeking is defined as a person’s intention to pursue assistance when experiencing psychological difficulties or feelings of distress (e.g., suicidal ideations; [95]). Mental health seeking stems from a person’s recognition that their psychological symptoms would be alleviated by assistance from others ([13]). The behavioral model of health service utilization emphasizes that individuals are influenced by micro- (e.g., actual and perceived health needs) and macro-level factors (e.g., social determinants of health) that influence proximally and distally a person’s decision to seek help ([5]). Individuals’ formal and informal networks are critical in the help-seeking process. Assistance with mental health may be sought through professional mental health providers or nonprofessional services such as family, friends, and online resources ([64]). However, help-seeking is affected by several barriers, including difficulty accessing support, lack of trust and fears about confidentiality, stigma, personal attitudes, feelings about disclosure, and a preference for informal support ([28]; [44]; [80]).

Culture is often overlooked and underexplored in a person’s decision to seek help ([61]; [85]). [79]’s ([79]) model examines the complex interaction between culture and facets of health decision-making, and highlights potential differences in perceptions of health, distress, and wellness that lead a person to seek help. Additionally, Saint Arnault’s model identifies the importance of a person’s social network as a driving mechanism for help-seeking, suggesting that the formulation of social capital (i.e., network size, trust, and knowledge of help-seeking pathways) occurs within a specific cultural context through a person’s social relationships. Experiences with acculturation add to this complexity by altering the potential resources for seeking assistance. For example, increases in acculturation are associated with increases in English proficiency ([91]) and greater knowledge of formal systems of assistance ([57]), which may facilitate help-seeking. Factors such as English language proficiency, often used to represent acculturation, are closely linked to service utilization ([32]), suggesting that greater acculturation may be associated with greater formal service use. However, Latinos/as often rely on informal networks (e.g., family, extended family, and friends) for support ([14]). Therefore, those individuals who are integrated into White American and Latinos/as networks may be best able to maximize their access to resources and seek assistance when needed.

### 1.5. The Present Study

The complex interplay between acculturation, social capital, help-seeking, and QoL requires new models to better describe Latino/a’s experiences and, by extension, to identify necessary support to enhance their lives. Figure 1 provides a conceptual framework depicting the association between study variables. For example, social influences (e.g., intercultural contact and relationships) are identified as factors affecting psychological acculturative experiences (i.e., cultural change and acculturative stress).

Moreover, the framework includes outcome variables indicative of adaptation, which are consistent with QoL. The framework aligns with existing stress process models (see [18]) and cross-cultural models (see [92]) that describe the temporal ordering of acculturation relative to these adaptive outcomes. Network and psychological acculturation then serve as cultural determinants for social capital, help-seeking, and QoL, further shaped by other determinants that influence health and health-related outcomes. The current study seeks to extend existing research by testing a structural model of quality of life that includes social networks as a determinant of psychological acculturation along with social capital and help-seeking. While studies examining acculturation and social capital are increasing, there is a dearth of research incorporating social networks as determinants of acculturation and other social and QoL outcomes for Latino/a adults. Moreover, a limited number of studies have examined the role of social capital in facilitating help-seeking and QoL. The current study tests the following hypotheses to address existing gaps in the literature:
**H_1_.** *Network acculturation will be negatively related to Latino/a enculturation and positively related to White American acculturation.*
**H_2_.** *Latino/a enculturation and White American acculturation will be positively related to social capital, as greater integration into diverse networks provides greater access to resources.*
**H_3_.** *Latino/a enculturation and White American acculturation will indirectly affect QoL through social capital and help-seeking but will not be directly related to Latino/a adults’ QoL.*
**H_4_.** *White American acculturation will be negatively related to help-seeking, while Latino/a enculturation will be positively related to help-seeking.*
**H_5_.** *Help-seeking will be positively related to QoL.*

## 2. Materials and Methods

### 2.1. Sample and Data Collection Procedures

The original survey design focused on the acculturation, negative social exchange networks, and mental health of Latino/a adults. However, a secondary analysis (e.g., examination of existing data to test a different set of hypotheses than initially intended) was conducted using cross-sectional egocentric social network data. Egocentric approaches ask a person (i.e., identified as ego) to list several people in their social network to describe potential network relationships. A crowdsourced population identifying as Latino/a/x or Hispanic between 18 and 65 years of age (*N* = 300), fluent in English, and living in the United States was recruited through Prolific^®^. Table 1 provides additional demographic information about the sample.

Prolific^®^ (Prolific Academic Ltd., London, UK) is a crowdsourcing platform that facilitates recruitment for research and employs state-of-the-art software for detecting bots that compromise data quality and study integrity. Prolific^®^ distributes the researcher’s recruiting message and a link to an electronic survey to Prolific^®^ members meeting the study’s criteria. Prolific^®^ recruits individuals from across the United States, making samples more representative than other convenience samples ([70]).

Prolific^®^ facilitates the study by monitoring survey completion and automatically identifying those who have timed out or are taking excessively long to finish the survey. All participants were compensated at a flat rate specified by Prolific^®^. Members are recruited until the desired sample size for the study is filled. The sample size for the structural equation model was calculated using the Analytics Calculator Index ([83]). A sample size of *N* = 177 was recommended, estimating a medium effect size for the model of 0.30, with six latent variables, 35 observed variables, a power of 0.80, and a *p*-value of 0.05. However, sample sizes between 200 and 300 are considered minimally sufficient to detect meaningful effects in structural equation models ([46]; [52]). Therefore, we recruited a sample of that size.

The construction of the survey and its implementation followed evidence-based practices outlined by the Tailored Design Method ([33]). This approach uses standardized recruitment methods and evidence-based survey practices (e.g., the content of the recruitment message, the cognitive flow of the survey content, and question types and placement) to minimize survey errors (e.g., nonresponse) and maximize recruitment and retention ([33]). All procedures were approved by the institutional review board associated with the study’s primary researcher.

### 2.2. Measures

#### 2.2.1. Generational Status

Generational status was once used as a proxy for acculturation and is still helpful in demographically understanding where people are situated in their family’s migration history ([72]; [87]). Participants were asked which of the following best described their generational status, with response options ranging from first to fourth generation. The following definitions were provided to participants. First-generation is defined as those participants who report nativity outside of the U.S. Those reporting second generation report nativity in the U.S. but have parents who report nativity outside of the U.S. Third-generation participants are those who are reporting nativity U.S. for themselves and their parents but report nativity outside of the U.S. for their grandparents. Lastly, fourth-generation participants report nativity in the U.S. for themselves, their parents, and their grandparents.

#### 2.2.2. Quality of Life

Several items were combined to measure Latinos/as’ quality of life. Two items from the WHOQOL-BREF ([96]) measuring a person’s satisfaction with their physical health and rating of their overall quality of life were used. These two items have demonstrated a strong relationship to the physical, psychological, social, and environmental domains of the WHOQOL-BREF ([82]). An additional item similar to physical health was also created that asked about respondents’ mental health. Response options for the items range from 1 (very dissatisfied/poor) to 5 (very satisfied/excellent). An exploratory factor analysis with these three items indicated a single factor explaining 69% of the variance with loading ranging from 0.81 to 0.84. While reliability and measurement error are closely related, certain forms of reliability or assessments of measurement error may be more pertinent ([65]). Composite reliability (CR) assesses whether the construct is reliable, with scores above 0.60 indicating adequate reliability ([2]; [37]). The CR for the current three-item scale is 0.96, suggesting strong reliability.

#### 2.2.3. Acculturation

##### Network Acculturation

Two measures were used in the current study to represent different aspects of acculturation. Name generators emphasize a person’s network’s structure (e.g., size), followed by additional questions (e.g., name interpreters) that provide information about the network’s composition. Using name generators and interpreters, network measures focusing on language use have been used to measure network aspects of acculturation and as a potential determinant of psychological acculturation ([7]; [12]). Therefore, the current study used name generators and name interpreters to assess different aspects of respondents’ social networks relative to acculturation. The current study placed greater emphasis on the composition of respondents’ networks, consistent with other network studies examining acculturation. These aspects have demonstrated a unique operationalization that is distinct from traditional psychological measures (see [12]). Using a name generator, respondents were asked to list individuals in their lives who were sources of stress. After listing the names, a piped text function in Qualtrics was used to create a follow-up question (i.e., name interpreter) asking respondents, “What language(s) do you speak with X?” with X representing the piped text name of the person identified by the respondent in the name generator. Response options ranged from 1 (all Spanish) to 5 (all English), with 3 representing equal use of Spanish and English. This question was used to create a network acculturation item of language use across all network members by summing and then mean-scoring items. Scores ranged from 1 (all Spanish) to 5 (all English), with higher scores indicating greater English use within one’s network. The CR for the network acculturation variable was 0.75, indicating an adequate level of reliability.

##### Enculturation and White American Acculturation

Additionally, a modified version of the Latino/a Acculturation Index ([8]) was used to measure Latino/a enculturation and White American acculturation. The LAI corresponds with traditional psychological measures in acculturation research, consisting of Latino/a enculturation and White American items measuring language use, identity and values, behaviors, and social relationships ([8]). A previous validation identified a shortened and full-length version of the LAI. However, neither of these versions of the scale demonstrated adequate model-data fit. The current study sought to identify the strongest measurement model of psychological acculturation using these items, which resulted in a five-item measure of Latino/a enculturation and a five-item measure of White American acculturation. Among the Latino/a enculturation items, Latino/a relationships, traditions, ethnic pride, and traditions were retained. Among White American acculturation items, items reflected preferences for their children’s identity, consumption of American food, friendships with White Americans, and preference for English music. Response options for the items ranged from 0 (*strongly disagree*) to 10 (*strongly agree*), with scores ranging from 0 to 50 for the two combined measures. Higher scores in the Latino/a enculturation and White American acculturation scales represent higher levels of enculturation and acculturation, respectively. The CR value for the modified version of the Latino/a enculturation scale was 0.86, while the CR value for the modified version of the White American acculturation scale was 0.76.

##### Social Capital

Several items (i.e., trust, social norms, and support network size) identified as useful measures of social capital ([11]; [67]) were used in the current study. The items “Most people can be trusted”, “I understand my social environment (e.g., the norms and ways)”, and “Overall, how many people could you contact if you need assistance (e.g., talk with someone, help you move, or give you a ride)?” were combined to create a latent social capital variable. Response options for the first two items ranged from 1 (strongly disagree) to 5 (strongly agree). Additionally, response options for the third item ranged from 1(0) to 7 (more than 5), with scores ranging from 3 to 17. Higher scores represent greater social capital. The CR for the current items was 0.70, indicating adequate reliability.

##### Help-Seeking

A modified version of the General Help Seeking Questionnaire ([31]) was used to measure a person’s intentions or willingness to seek assistance when experiencing psychological distress (i.e., personal or emotional problems or suicidal ideations). The modified version of the GHSQ consisted of 18 items asking about a respondent’s likelihood of seeking assistance from different informal (e.g., friends and family) and formal (e.g., mental health professional, phone/crisis line, and doctor/general practitioner) sources ([95]). Response options for the scale items ranged from 1 (extremely unlikely) to 7 (extremely likely). While the complete scale was initially used, the item regarding other sources of help-seeking was missing a substantial amount of data and was removed from analyses. Additionally, the item asking about help negation was negatively related to other items in the measurement model, resulting in poor model fit. Negation items in other studies have been removed from analyses (see [95]), which was done in the current study, given the items’ effect on model fit. As a result, these items were removed from the analyses.

The emergence of computer-based and app-based sources of assistance prompted the inclusion of an additional item in each scale (see [25]). The new item asked respondents about the likelihood of seeking help online or using a mental health app on their smartphone. Response options provided were consistent with the original scale. Scale scores for the modified version of the scale ranged from 18 to 126, with higher scores indicating a greater likelihood of help-seeking. The factor loading for the item in the general psychological help-seeking scale was 0.42 and 0.69 in the scale asking about suicidal ideations, producing comparable loadings to other items in their respective scales. The CR values for the general psychological help-seeking and suicidal ideations scales were 0.96 and 0.97, respectively, while the overall scale produced a CR of 0.95.

### 2.3. Analytic Plan

SPSS 29 and AMOS 29 were used to conduct analyses to test study hypotheses. Less than 1% of data were missing in the variables included for analyses. A median value replacement strategy was used to address the missing values in the data set (see [100]). Univariate analyses (e.g., measures of central tendency and patterns of responses) were used to describe acculturation, social capital, help-seeking, and QoL responses. Pearson’s correlations were computed to test initial associations between variables. Structural equation modeling was used to test the measurement models for latent variables and a conceptual model. Demographic variables may be important controls. However, they may add noise when they are not the focus of analyses or inaccurately provide control when insufficiently measured ([63]). Given that other study variables were the focus, demographic variables were not included in the structural equation model, including generational status, which often duplicates acculturation measures.

Measurement models identify the observed variables relative to a latent construct that delineates a proposed theoretical model, while the structural model tests hypothesized relationships between latent constructs that describe conceptual associations ([53]). While no single index indicates appropriate model fit, several indices used collectively can point to the appropriateness of the model ([78]). First, a chi-square to degrees of freedom ratio provides an initial assessment of the model, with ratios less than 2.00 and 3.00 indicating acceptable model fit ([1]). The normed fit index (NFI), relative fit index (RFI), incremental fit index (IFI), Tucker–Lewis Index (TLI), and comparative fit index (CFI) were used to assess fit. Values of 0.90 or larger suggest a reasonable model fit ([52]; [24]), while values larger than 0.95 indicate a more robust model ([22]; [47]).

The root mean square error of approximation (RMSEA) relaxes the idea of a perfect model and assesses approximal fit ([78]). Values less than 0.05 suggest a close fit, while values below 0.08 indicate an acceptable fit ([99]; [48]). The standardized root mean square residual (SRMR) measures the discrepancy in fit between covariance matrices, with values below 0.08 suggesting an acceptable fit ([56]; [47]). Modification indices were used to add paths between error variances within measures or between theoretically relevant constructs. For error variances, only paths within subscales were added, given the similarity between item content and to retain the conceptual distinction between the subscales. There are a number of pathways through which acculturation and social capital likely relate to QoL outcomes among Latinos/as. Therefore, several paths between acculturation, social capital, help-seeking, and QoL were examined. The estimands function in AMOS was used to specify which direct and indirect paths and sources of serial mediation would be tested in the model. Bootstrapping was used in AMOS to generate estimates and *p*-values for direct and indirect effects in the model.

## 3. Results

### 3.1. Univariate Analysis

Table 2 provides the means and standard deviations for study variables. The average QoL score in the sample was 10.10 (*SD* = 2.51) out of a possible score of 15. Network language use score, representing network acculturation, averaged 4.01 (*SD* = 1.02), indicating a higher level of English use. Latino/a enculturation and White American acculturation scores were 31.93 (*SD* = 11.76) and 25.80 (*SD* = 6.78) out of a possible 50. The sample’s mean social capital level was 11.15 (*SD* = 2.55) out of a possible 17. To further contextualize study variables, QoL, enculturation, White American acculturation, social capital, and help-seeking were examined descriptively by generational status, a proxy for acculturation. Figure 2 provides the means for each variable by generational status group.

### 3.2. Bivariate Analyses

Network acculturation was negatively correlated with Latino/a enculturation (*r* = −0.47, *p* < 0.001) and positively correlated with White American acculturation (*r* = 0.25, *p* < 0.001), but was not correlated with any other study variables. Table 2 provides the correlations between study variables. Latino/a enculturation was positively related to social capital (*r* = 0.12, *p =* 0.018), help-seeking (*r* = 0.25, *p* < 0.001), and QoL (*r* = 0.16, *p* = 0.002), suggesting that greater emersion into Latino/a culture is associated with several health benefits. White American acculturation demonstrated a similar relationship to social capital (*r* = 0.21, *p* < 0.001), help-seeking (*r* = 0.18, *p* = 0.001), and QoL (*r* = 0.14, *p* = 0.007); however, White American acculturation was more strongly related to social capital and weakly related to help-seeking compared to Latino/a enculturation. Social capital was moderately related to help-seeking. At the bivariate level, help-seeking was positively related to QoL (*r* = 0.34, *p* < 0.001).

### 3.3. Structural Equation Modeling

Several goodness of fit indices were examined to assess the fit of the data to the proposed measurement model. Figure 3 provides a structural evaluation model examining study variables with standard estimates for main effects. The chi-square to degrees of freedom ratio (χ^2^/df = 967/498 = 1.94) was below 2.00, suggesting a good-fitting model.

The CFI (0.92), TLI (0.90), and IFI (0.92) were equal to or larger than 0.90, meeting the more liberal threshold. However, the NFI (0.85) and RFI (0.81) were all below 0.90. Similarly, RMSEA (0.06) was slightly below the more liberal threshold of 0.08. Lastly, the SRMR for the model was 0.08.

Both direct and indirect effects were examined in the structural equation model. Table 3 provides the direct effects examining network acculturation, acculturation, social capital, and help-seeking on Latino/a adults’ quality of life. Network acculturation was directly related to Latino/a adults’ enculturation (*p* = 0.001) and acculturation (*p* = 0.007). However, network acculturation demonstrated a stronger negative relationship with Latino/a enculturation than its positive relationship with White American acculturation, indicating that network language use associated with social relationships may play a more prominent role in the maintenance of Latino/a culture. Additionally, Latino/a enculturation (*p* = 0.002) and White American acculturation demonstrated (*p* = 0.003) a direct relationship with social capital, each contributing equally to accumulating social resources. Of the acculturation-related variables, only Latino/a enculturation (*p* = 0.029) was significantly related to help-seeking, with higher Latino/a enculturation associated with greater intentions to seek help. Social capital demonstrated a direct relationship with both help-seeking (*p* = 0.004) and quality of life (*p* < 0.003), with higher social capital associated with greater intention to seek help, along with greater QoL.

Several indirect effects and serial mediation were observed in the model. Table 4 provides standardized estimates, standard errors, and *p*-values for the indirect effects and serial mediation tested in the current study. Network acculturation demonstrated a statistically significant indirect effect on social capital through Latino/a enculturation (*p* = 0.001) and White American acculturation (*p* = 0.002), suggesting that network acculturation plays a role in maintaining and generating network resources. Multiple paths between network acculturation and help-seeking were examined. Pathways through Latino/a enculturation (*p* = 0.019) and serial mediation by enculturation and social capital (*p* = 0.001) were significant. Similarly, the relationship between network acculturation and help-seeking was also serially mediated through White American acculturation and social capital (*p* = 0.001), identifying multiple pathways through which network acculturation aids in accumulating resources that facilitate mental health help-seeking.

The indirect effects of Latino/a enculturation and White American acculturation were examined independently of network acculturation. Latino/a enculturation (*p* = 0.001) and White American acculturation (*p* = 0.003) demonstrated a statistically significant effect on help-seeking through social capital. Indirect effects and serial mediation related to QoL were also examined. Network acculturation was significantly related to QoL through Latino/a enculturation and social capital (*p* = 0.002). Similarly, network acculturation was significantly related to QoL through White American acculturation and social capital (*p* = 0.003), identifying multiple pathways through which different aspects of acculturation were associated with QoL. Additionally, Latino/a enculturation (*p* = 0.003) and White American acculturation (*p* = 0.004) demonstrated significant indirect effects on QoL through social capital independent of network acculturation, indicating that psychological acculturation alone was also a driving factor associated with one’s QoL. No indirect path or serial mediation involving help-seeking was significantly associated with QoL.

## 4. Discussion

The current study investigated the roles of acculturation and social capital as determinants of help-seeking and QoL. Findings from the study make an important contribution to the acculturation literature by revealing the pathways through which network-related factors likely contribute to Latino/a’s decisions to seek help and facilitate access to social capital. Additionally, the current study is one of few studies that focuses on the contribution of negative relationships to a person’s psychological acculturative experiences. While network acculturation was not directly related to these outcomes, a thorough examination of indirect pathways and serial mediation across variables exposed a complex interplay between these factors, help-seeking, and QoL. While only Latino/a enculturation was directly related to help-seeking, enculturation and acculturation demonstrated indirect effects through social capital. However, help-seeking was not related to Latinos/as’ QoL, likely associated with the psychological difficulties and distress that drive help-seeking intentions.

Findings from the current study add depth to what is known about the relationship between network acculturation and psychological acculturation. Network acculturation demonstrated a stronger negative relationship to Latino/a enculturation compared to its positive relationship with White American acculturation. This suggests that knowing and using English in one’s social networks is more strongly related to whether one identifies with Latinx culture than White American culture. Maintaining one’s heritage language takes effort and commitment, particularly in communities where the language is not widely spoken. The lack of incorporation of Spanish into a person’s conversations and communications with others is connected to their integration into Latino/a culture.

The influence of network-derived acculturation aligns with stress-based acculturation models that identify group-level factors as determinants of a person’s acculturation ([18]; [92]). Here, network acculturation demonstrated a significant negative indirect effect on social capital and QoL through Latino/a enculturation. Interestingly, the opposite was observed through White American acculturation. That is, increases in network acculturation (i.e., English use across one’s network) were associated with greater social capital and quality of life for those more strongly identifying with White American culture. This suggests that those who more strongly identify with Latino/a culture but have a largely English-speaking social network would have less access to needed resources, which results in lower QoL. Conversely, those more strongly embracing White American culture with a largely English-speaking social network would experience greater integration and, therefore, greater access to social capital and quality of life. This finding highlights the role that social networks (i.e., how individuals are connected) play in how individuals acculturate and the importance of cultural fit and serves as a bridge to aspects of acculturation research emphasizing interaction within (i.e., intercultural) and between (i.e., intercultural) groups (e.g., pressure to acculturate).

As in other studies ([90]; [71]), experiences with acculturation were related to a person’s social capital. Access to social capital is driven by group membership across several networks (e.g., family, friends, or coworkers), which is contingent on different characteristics seen as central to the group, including culture and race ([62]). [20]’s ([20]) notion of cultural capital provides a bridge for understanding the role of acculturation. Although initially applied in the context of social class, cultural capital argues that cultural knowledge, behaviors, and values serve as substrates for building relationships that afford demonstrators particular privileges within networks. In the current study, individuals who were integrated into Latino/a and White American communities had the greatest amount of social capital, suggesting that integration may be the best acculturation strategy for maximizing potential support. This is consistent with what is observed above, which suggests that a potential lack of cultural congruence with one’s network may result in lower social capital that influences a person’s QoL.

The psychological (e.g., self-esteem) and social (e.g., access to social support) benefits of belonging are well-established in this literature ([51]; [81]). Cultural similarities likely facilitate membership in different networks by providing a common language, shared values, and engagement in similar activities (e.g., attending the same religious service) that build trust and a sense of reciprocity, and facilitate integration. However, Latino/a enculturation and White American acculturation were not uniformly related to help-seeking in the current study. Latino/a enculturation was directly related to help-seeking, but also demonstrated an indirect effect through social capital. This suggests that integration into Latino/a culture is important for generating resources and increasing a person’s intentions for seeking help. Conversely, White American acculturation was only indirectly related to help-seeking through social capital, indicating that it may be important in accessing needed resources, but not in directly seeking help. Latinos/as’ supportive networks are largely composed of family and friends ([12]), and therefore, cultural alignment likely strengthens ties in culturally homogenous networks and facilitates closer relationships that provide avenues for seeking assistance.

Interestingly, help-seeking was unrelated to QoL. A person’s intentions to seek help are associated with symptoms adversely affecting their life ([13]). It is likely that individuals who are seeking assistance are experiencing psychological distress or need additional resources to manage their symptoms. While alleviating psychological distress and addressing resource gaps is critical, it may not translate to changes in QoL. A distinction has been made in the positive psychology literature that differentiates mental illness from mental health, suggesting that the alleviation of illness does not always translate to improved health ([51]). For example, a person may not be satisfied by the improvements or changes in health associated with their help-seeking.

In the current study, social capital was the strongest predictor of QoL. Social capital has been a focus of public health and social epidemiological research for the last 20 years because of its connection to physical and mental health ([26]; [35]; [66]). Individuals with higher social capital have more extensive social networks that provide access to resources ([20]; [55]). In the current study, social capital was a predictor of help-seeking. However, help-seeking was unrelated to QoL, suggesting that the benefits gleaned from social capital that enhance QoL may be unrelated to resources that help manage psychological distress. Social integration has been thought to yield direct benefits that enhance a person’s life that are distinct from other resources generated ([21]). A more recent theoretical approach has suggested that social relationships are not simply determinants of health but reflect a distinct aspect of one’s health that contributes to a tripartite model of overall wellness ([34]). Poorer social capital may be indicative of strained relationships that are stress-evoking ([9]), while positive relationships may reflect a more robust network capable of producing social health benefits.

Several limitations are worthy of consideration in relation to the findings. First, the fit indices did not reach the anticipated thresholds, suggesting a need to improve the fit of the model. The lack of fit may be associated with several issues. The network measure of acculturation is limited to language use. Increasing the types of network acculturation measures may improve the measurement model for network acculturation and improve the overall model fit. Additionally, an English-speaking population was the focus of the study, which ultimately limited the acculturative experiences included in the analyses. While the AVE and CR were appropriate, the use of an English-speaking sample does pose a threat to the reliability and validity of measures. A more robust representation of acculturative experience will improve measurement and better model people’s experiences of acculturation. Future studies should expand network items to include different aspects of acculturation to determine whether other network indicators demonstrate a significant relationship with psychological acculturation and health outcomes. Moreover, a more linguistically diverse sample should be included to capture the range of acculturative experiences fully and further explore the meaningfulness of network variables. The items used to represent social capital are limited and largely represent personal social capital. Social capital is a complex construct that includes personal and ecological measures. Similarly, QoL represents numerous facets of a person’s life, requiring more extensive measures to best understand the impacts of these study variables on Latinos/a’s lives. Lastly, help-seeking was found to be unrelated to QoL. Help-seeking in the current study was measured by one’s intention to seek assistance, which differs from a person’s actual service use. Therefore, the intention to seek help was used as a proxy for help-seeking. Future research should include a person’s intention to seek help along with objective measures of help-seeking to determine whether help-seeking intentions are associated with QoL through one’s actual service use. Therefore, more robust measures are needed to further assess the relationships between variables examined in the model.

## 5. Conclusions

As health and quality of life models continue to improve in their descriptive abilities, acculturation becomes further situated as a key factor in accessing assistance and resources that are critical to Latino/a adults. Including network acculturation in research highlights its role as a social determinant of psychological acculturation experiences and other important outcomes that drive life quality. Network indicators of acculturation need to be expanded to include factors relevant to other acculturation domains beyond language (e.g., social relationships, cultural practices and behaviors, and values and beliefs) to understand whether more complete measures of network acculturation behave as anticipated. Future researchers should continue to explore determinants of acculturation, along with the role of acculturation as a predictor of health outcomes, to understand the complex pathways through which acculturative factors relate to health and QoL.

## Figures and Tables

**Figure 1 behavsci-15-00388-f001:**
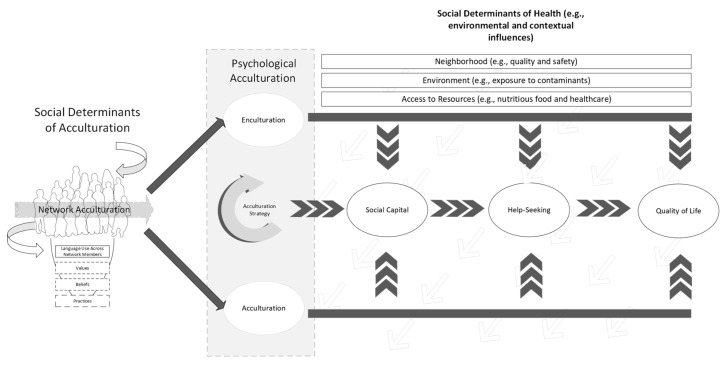
Conceptual model depicting the associations between network and psychological acculturation and social capital, help-seeking, and QoL.

**Figure 2 behavsci-15-00388-f002:**
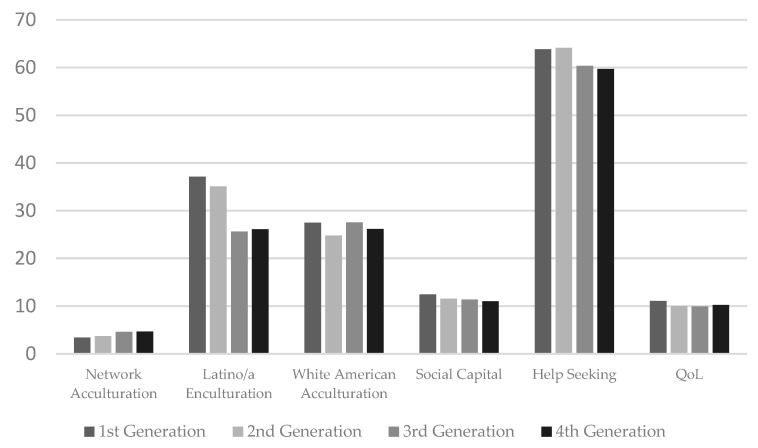
Means of network acculturation, Latino/a enculturation, White American acculturation, social capital, help-seeking, and QoL by generational status group.

**Figure 3 behavsci-15-00388-f003:**
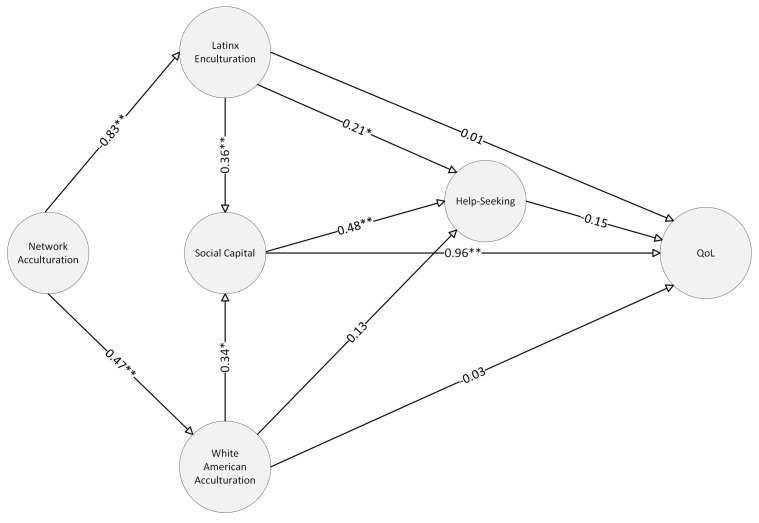
Structural model examining network and psychological acculturation, social capital, and help-seeking on quality of life (QoL). * *p* < 0.05; ** *p* < 0.01.

**Table 1 behavsci-15-00388-t001:** Demographic characteristics of the sample (*N* = 300).

	*M/%*
Age	
Gender	
Men	43.3%
Women	56.7%
	100.00%
Generational Status	
1st	8.3%
2nd	55.0%
3rd	23.0%
4th	13.3%
Missing	0.4%
	100%
Relationship Status	
Single	47.7%
Married	26.3%
Divorced	1.0%
Living with Partner	11.7%
In a Committed Relationship	13.3%
	100.00%
Highest Level of Education	
Less than High School	1.7%
High School Diploma	16.7%
Vocational Training	3.7%
Associate’s degree	10.7%
Some College	20.0%
Bachelor’s Degree	34.3%
Master’s Degree	10.3%
PhD, MD, JD. Or other doctorate	1.7%
Missing	0.90%
	100%

**Table 2 behavsci-15-00388-t002:** Means, standard deviations, and Pearson’s correlations between study variables.

	1	2	3	4	5	6
Network Acculturation	1					
2. Latino/a Enculturation	−0.47 ***	1				
3. WA Acculturation	0.25 ***	−0.17 **	1			
4. Social Capital	−0.04	0.12 *	0.21 ***	1		
5. Help-Seeking	−0.04	0.25 ***	0.18 **	0.36 ***	1	
6. QoL	−0.04	0.16 **	0.14 **	0.53 ***	0.34 ***	1
*M*	4.013	31.93	25.80	11.15	62.67	10.10
*SD*	1.02	11.76	6.78	2.55	23.27	2.51

* *p* < 0.05; ** *p* < 0.01; *** *p* < 0.001.

**Table 3 behavsci-15-00388-t003:** Direct effects of network acculturation, Latino/a enculturation, White American acculturation, social capital, and help-seeking on QoL.

	Standardized Estimate	S.E.	Critical Ratio	*p* Value
NA→Latino/a Enculturation	−0.83	0.69	−3.38	0.002 **
NA→WA Acculturation	0.47	0.18	2.71	0.003 **
Latino/a Enculturation→Social Capital	0.36	0.04	3.21	0.002 **
WA Acculturation→Social Capital	0.35	0.15	2.28	0.003 *
Social capital→Help-Seeking	0.48	0.26	3.98	0.004 **
Latino/a Enculturation→Help-Seeking	0.21	0.05	2.67	0.029 **
WA Acculturation→Help-Seeking	0.13	0.16	1.42	0.226
Social Capital→QoL	0.96	0.35	4.07	0.003 **
Help-Seeking→QoL	−0.15	0.12	−0.70	0.277
Latino/a Enculturation→QoL	0.01	0.05	−0.03	0.923
WA Acculturation→QoL	−0.04	0.15	−0.32	0.821

* *p* < 0.05; ** *p* < 0.01; NA = network acculturation; WA = White American.

**Table 4 behavsci-15-00388-t004:** Indirect effects and serial mediation between study variables.

Indirect Paths/Serial Mediation	Standardized Estimate	S.E.	*p* Value
NA→Latino/a Enculturation→Social Capital	−0.29	0.15	0.001 **
NA→Latino/a Enculturation→Help-Seeking	−0.29	0.21	0.019 *
NA→Latino/a Enculturation→Social Capital→Help-Seeking	−0.23	0.16	0.001 **
NA→Latino/a Enculturation→Social Capital→QoL	−0.43	0.31	0.002 **
NA→Latino/a Enculturation→QoL	−0.01	0.17	0.911
NA→Latino/a Enculturation→Help-Seeking→QoL	0.04	0.05	0.143
NA→Latino/a Enculturation→Social Capital→Help-seeking→QoL	0.03	0.10	0.203
NA→WA Acculturation→QoL	−0.02	0.09	0.836
NA→WA Acculturation→Social Capital	0.16	0.07	0.002 **
NA→WA Acculturation→Social Capital→Help-Seeking	0.13	0.08	0.001 **
NA→WA Acculturation→Help-Seeking	0.10	0.08	0.161
NA→WA Acculturation→Social Capital→QoL	0.24	0.13	0.003 **
NA→WA Acculturation→Help-Seeking→QoL	−0.01	0.02	0.288
NA→WA Acculturation→Social Capital→Help-Seeking→QoL	−0.02	0.04	0.398
Latino/a Enculturation→Social Capital→Help-Seeking	0.10	0.05	0.001 **
Latino/a Enculturation→Social Capital→QoL	0.19	0.09	0.003 **
Latino/a Enculturation→Help-Seeking→QoL	−0.02	0.02	0.192
Latino/a Enculturation→Social Capital→Help-Seeking→QoL	−0.02	0.03	0.212
WA Acculturation→Social Capital→Help-Seeking	0.27	0.23	0.003 **
WA Acculturation→Social Capital→QoL	0.50	0.39	0.004 **
WA Acculturation→Help-Seeking→QoL	−0.01	0.05	0.327
WA Acculturation→Social Capital→Help-Seeking→QoL	−0.02	0.10	0.398
Social Capital→Help-Seeking→QoL	−0.10	0.27	0.249

* *p* < 0.05; ** *p* < 0.01; NA = network acculturation; WA = White America.

## Data Availability

Data associated with the current study are available upon reasonable request and in accordance with the granting institutional review board’s rules and regulations.

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
