# Peer review of "Quality of Life Among Latino/a Adults: Examining the Serial Mediation of Network Acculturation, Psychological Acculturation, Social Capital, and Helping-Seeking"

_behavsci, 2025, doi:10.3390/bs15030388_

Round 1
Reviewer 1 Report
Comments and Suggestions for Authors
This article is a secondary data analysis of 300 data points derived from a crowdsourced population identifying of Latinos ages 18 and 65 years of age and are fluent in English. Given that language has one of the most predictive values for the acculturation process, it was hard to understand how achieved sample is representative of the wide spectrum of acculturation experience for Latinos in the United States. Though this could be mitigated somewhat by including this issue as a limitation by the authors, I think the entire manuscript could be strengthen if more transparency is lent to this issue from the title (i.e., English dominant Latinos) to the interpretation and discussion of the findings. The authors seem to reference a lot of their own work and that of Berry et al. throughout the manuscript, which must be improved since a lot of the health research on this topic seemed to be missing.
Introduction
· Given that the specific hypothesis of the paper uses the term “White American” I think it will be important for the authors to unpack this a bit more in this section, especially in the section on acculturation that cites a lot of the work by Berry who actually used the term “Anglo” to describe the dominant group in the US context. Some sociologists actually use WASP more broadly to include all White Protestant Americans of Northwestern European and Northern European ancestry, which again paints a more precise cultural and social portrait of the comparison group.
· Here and throughout the authors use the term Latinx to describe their sample. Did the authors ask participants whether this was their preferred term? At the very least a rationale should be provided, since there is increasing evidence that the vast majority of individuals who self-identify in this group do not use the Latinx term. Two excellent papers on this are shared below:
o Borrell LN, Echeverria SE. The use of Latinx in public health research when referencing Hispanic or Latino populations. Soc Sci Med. Jun 2022;302:114977. doi:10.1016/j.socscimed.2022.114977
o Viladrich, A. Latinx for whom? Reflections upon the Linguistic Shaping of Latin American Identities in the United States Ethnic and Racial Studies, 2024
Methods
· The name generator that was used to derive the acculturation network-level measure solely focused on individuals in the lives of respondents who were sources of “stress.” Typically, social network generators are more concerned in identifying the alters with high frequency of contact or degree of closeness to the ego. Can the authors explain their theoretical and methodological rationale for concentrating on only stress-inducing alters? This coupled with the sampling of only English dominant Latinos only as noted below bring up issues of internal and external validity that are important to flesh out in this section
Results
· I appreciate that the authors used structural equation modeling to test their 5 hypotheses, but it was unclear why certain they did not include standard SES-related covariates such as gender, age, and income. All would fit the criteria of confounders for the relationships of interest.
· Given that the authors are performing multiple hypothesis tests simultaneously, why did they not apply the Bonferroni correction?
Discussion
· I did not follow this sentence in the conclusion “As models continue to advance, acculturation becomes further situated as a key factor in accessing assistance and resources that are critical to the health and well-being of Latinx adults.” Can the authors please clarify what they mean by models continuing to advance? Also, this seems like an important opportunity for the authors to share with readers what are the next steps they deem important in acculturation research that focuses on network-level processes especially in light of the null results their approach yielded
Author Response
We want to thank the reviewers for their comments. We hope that they find that we have addressed their concern or provided a suitable response that answers their questions. Ultimately, we believe that their contribution has strengthened the quality of the manuscript. We look forward to your response.
This article is a secondary data analysis of 300 data points derived from a crowdsourced population identifying of Latinos ages 18 and 65 years of age and are fluent in English. Given that language has one of the most predictive values for the acculturation process, it was hard to understand how achieved sample is representative of the wide spectrum of acculturation experience for Latinos in the United States. Though this could be mitigated somewhat by including this issue as a limitation by the authors, I think the entire manuscript could be strengthen if more transparency is lent to this issue from the title (i.e., English dominant Latinos) to the interpretation and discussion of the findings.
- (p. 17, lines 288-293) We agree with the reviewers. The representation of full acculturation experiences is limited. However, the opposite is also true. While language is an important indicator, there are other indicators of acculturation. The sample is not as limiting as one might expect. Moreover, this is often not a criticism levied against samples that are purely first-generation or only Spanish speakers, despite that the same limitations apply. Because of the cross-sectional nature of acculturation research, all forms of research are needed to present acculturative experiences so that data can be aggregated across studies to present a fuller picture. We understand the concern and have noted the limitation in the limitation section to increase transparency
The authors seem to reference a lot of their own work and that of Berry et al. throughout the manuscript, which must be improved since a lot of the health research on this topic seemed to be missing.
- We have reduced the number of self-citations and included more from the broader literature. We have completed a significant amount of work in the area of acculturation and networks and had to retain some of those citations.
Introduction · Given that the specific hypothesis of the paper uses the term “White American” I think it will be important for the authors to unpack this a bit more in this section, especially in the section on acculturation that cites a lot of the work by Berry who actually used the term “Anglo” to describe the dominant group in the US context. Some sociologists actually use WASP more broadly to include all White Protestant Americans of Northwestern European and Northern European ancestry, which again paints a more precise cultural and social portrait of the comparison group.
- In the Southwestern United States, the term Anglo is used more frequently, but people outside of that region do not use the term. In the United States broadly, Anglo is not used as a racial designation. In the US census, non-Hispanic White is used. We added “American” based on our previous work. European immigrants completing surveys in the past have reported that White alone might capture those who were first-generation and potentially skew the results. We also have a portion of the White population that is Catholic, Jewish, or another religious group that is not Protestant. Berry is Canadian and is not a US researcher. A portion of his work was focused on language, which he differentiated as Anglophones and Francophones, hence the use of the term Anglo. We have developed several measures and conducted numerous studies using the term White American and have not encountered any problems with its use.
Here and throughout the authors use the term Latinx to describe their sample. Did the authors ask participants whether this was their preferred term? At the very least a rationale should be provided, since there is increasing evidence that the vast majority of individuals who self-identify in this group do not use the Latinx term. Two excellent papers on this are shared below: - - or
Borrell LN, Echeverria SE. The use of Latinx in public health research when referencing Hispanic Latino populations. doi:10.1016/j.socscimed.2022.114977 Soc Sci Med. Jun 2022;302:114977.
Viladrich, A. Latinx for whom? Reflections upon the Linguistic Shaping of Latin American Identities in the United States Ethnic and Racial Studies, 2024 Methods ·
- I’m Latino, and understand the controversy well. I appreciate the references. The use of the term is intended to prompt a broader level of inclusiveness. Yes, there is research that indicates that only a few individuals have embraced the term. However, the same can be said for Latino/a. A more significant portion of people prefer the term Hispanic over Latino/a. Hispanic was an ascribed identity and Latino/a, and Latinx are terms that move away from that ascribed identity. In some circumstances, people have moved to Latine, which is more consistent with Spanish. While our inclusion criteria included all identities, the acculturation measure only specified Latino/a and Latinx. As such, we cannot use Latine. However, we have opted to use Latino/a in the manuscript so that terminology is more representative of the larger group.
- The name generator that was used to derive the acculturation network-level measure solely focused on individuals in the lives of respondents who were sources of “stress.” Typically, social network generators are more concerned in identifying the alters with high frequency of contact or degree of closeness to the ego. Can the authors explain their theoretical and methodological rationale for concentrating on only stress-inducing alters?
- Yes, the reviewer is correct that we have turned that methodology and used it in another way. We believe there is substantial support for its use and a substantial gap in understanding the influence of negative relationships. Relationship models (e.g., social convoy model), have specified that both negative and positive relationships are critical in understanding the full impact of relationships on different outcomes. The name generator method is often used in such studies to gather information on positive relationships but is not used to understand the effect of negative relationships. Additionally, there is a considerable amount of research that suggests that social capital, and the qualities of positive relationships, may also exist with negative relationships (e.g., relationships tied to negative activity or those associated with negative social exchange), often described as dark social capital. While we did gather social capital data associated with the boarder study, it is not part of the current analysis. We have also published work demonstrating the association between these measures and standardized measures used to screen for mental health. Moreover, in other work, we have compared the quality of relationships often associated with social capital (e.g., frequency, relationship quality, and trust) between positive and negative networks. Across studies, we have found that measures generated from this methodology across network types behave as anticipated (e.g., demonstrate construct or criterion validity with single-item indicators, known standardized scales, and AVE and CR scores).
This coupled with the sampling of only English dominant Latinos only as noted below bring up issues of internal and external validity that are important to flesh out in this section.
- We agree that an English dominant population can be problematic. While the study was conducted in English, the language items related to Spanish fluency demonstrated variation, suggesting that the study is more representative of acculturation experiences that the English participation would suggest. Additionally, studies using the language of completion as a proxy of acculturation have been criticized as not fully representing acculturation. However, the indices indicating the reliability and validity were addressed. We examined the AVE and CR for each existing measurement model. Actually, in doing so, we have provided a more robust test of the internal and external validity than is usually provided by other studies, which rely on alpha and previous validation efforts. Our study demonstrates that the measures included meet appropriate standards. However, we have added a statement to the limitations section reflecting this concern.
Results · I appreciate that the authors used structural equation modeling to test their 5 hypotheses, but it was unclear why certain they did not include standard SES-related covariates such as gender, age, and income. All would fit the criteria of confounders for the relationships of interest.
- There are several rationales. First, it was not the focus of the study. The inclusion of these variables would have led to additional parameters and additional hypotheses. Second, as the reviewers have noted, there were several variables, so we had to be mindful of the power associated with the statistical tests. The inclusion of additional variables added exponentially to the number of parameters included for analysis. While the variables may be important controls, some have suggested that the inclusion of variables in an SEM model, where they are not the focus, may add noise to the model (Memon et al., 2024). We did provide this rationale in the analytic plan section.
Given that the authors are performing multiple hypothesis tests simultaneously, why did they not apply the Bonferroni correction?
- I have never seen anyone report a Bonferroni correction for SEM. SEM is robust to multiple tests, which is why we are using SEM. There are no post-hoc comparisons. It is the same with multiple regression. Each t-test in the model is a test, but you don’t use a Bonferroni correction for a single model; you do it for multiple regression models. If this was the case, you would never be able to run any SEM with sizes less than several 1000 because the measurement model alone would constitute multiple tests. I have searched for papers on the topic, and nothing is suggested beyond a manual correction report on Research Gate and only when requested by a reviewer. I don’t believe that a correction is necessary given the statistical test is robust to testing multiple parameters within a single model. We would also argue that is the value of the fit indices for the overall model, rather than relying on chi-square and other such indicators of statistical significance. Ours were suggestive of improvements, which we emphasized in the limitation as part of the response.
Discussion · I did not follow this sentence in the conclusion “As models continue to advance, acculturation becomes further situated as a key factor in accessing assistance and resources that are critical to the health and well-being of Latinx adults.” Can the authors please clarify what they mean by models continuing to advance?
- All we meant is that acculturation models continue to develop and improve in their ability to describe people’s experiences. Models that have been used for years have extremely limited conceptualization of an incredibly complex process. Models must advance (i.e., continue to improve) to describe acculturation in ways that reflect its true complexity.
Also, this seems like an important opportunity for the authors to share with readers what are the next steps they deem important in acculturation research that focuses on network-level processes especially in light of the null results their approach yielded.
- We thank the reviewers for their suggestions. We have added content in the discussion to reflect this need.
- (p. 17, lines 715-721) However, the complex roles that network acculturation maintains require further investigation in light of this unanticipated finding (i.e., positive relationship to acculturation). Network indicators of acculturation need to be expanded to include factors relevant to other acculturation domains beyond language (e.g., social relationships, cultural practices and behaviors, and values and beliefs) to understand whether more complete measures of network acculturation behave as anticipated. Future researchers should continue to explore determinants of acculturation, along with the role of acculturation as a predictor of health outcomes, to understand the complex pathways through which acculturative factors relate to health and QoL.
Reviewer 2 Report
Comments and Suggestions for Authors
This is a mostly well-done manuscript with interesting and relevant findings; several portions are excellent (much of discussion, summary of previous lit); the method is solid. My main two recommendations are for more clarity in wording at some places (I note below all of the places that I saw) and, more importantly, a clearer connection of the review of literature to the hypotheses posed, so that there is a clear argument for why you expect what you do in each hypothesis.
The manuscript is mostly very well written. Normally, I have two lists of comments—editorial and substantive. But there are so few editorial comments, I will just list all comments below by line number.
- Solid abstract—quite clear
- Introduction introduces key concepts clearly, with good definitions and clear documentation in previous research.
- 88: I was a little unsure what you mean by “the cascading of effects…early in life”. The use of “the” suggests you’ve introduced or talked about such a cascade before.
- 91-92: “acculturation…assimilation.” The terms are used as if they have already been defined. Without prior context of some sort, the reader may not know how you are using these seemingly similar terms (i.e., are these different things?). I had a question about the next sentence (“However…” and how it related, but definitions of the terms, as I think you are using them, would make the “however” sentence clear.
- 102: “despite their diverse origins and applications.” I’m not sure what you mean. Are you beginning a list of description of different types of adjustment with “psychological acculturation”? If so, preview the list so the reader knows where the argument is going. Is this distinction your own (as it seems from the writing), or is this based on Berry’s well-known distinctions? If the list of types of acculturation is your own, use language to suggest that (e.g., various sources suggest three main types of acculturation relevant to this study,” or something like that).
- 114: “Social relationships are ore to how acculturation is described and understood.” Provide evidence for this claim. I know that years ago, some scholars in my own field (communication) said that ability to form close relationships was one of the main aspects of “intercultural communication competence” (Hammer, Gudykunst, & Wiseman, 1978), but you probably have your own sources. The current evidence (Rodriguez et al., 2002, etc.) is about measures that scholars have introduced, which doesn’t establish the connection between social relationships and acculturation that findings would
- 122-123: “with whom a person shares or diverts from culturally”: I’m not sure what you mean by “diverts” here, so maybe another word. And because of the structure of the sentence, the section action has two prepositions (with whom diverts from)à “…social relationships in which a person approaches or moves away from others culturally…”?
- 127: “For example…strategies.” Again, strategies are mention that have not been previously outlined, so they lack context.
- Note: There is a lot of self-citation here.
- 157-158: “social capital . . . identifies the pathways…”: unusual wording
- 160: “social capital has demonstrated inconsistent results”—in terms of what?
- 166-167: “increased exposure to American culture changes [add ‘s’] important elements..—which elements?
- 189: Does “it” here refer to culture or to Arnault’s model?
- 194, 200: “enculturation…separation…acculturation…” Terms need definition or introduction, probably above in the paper,.
- 225: Probably add “A” before “network measure” (H1). But since later Hs are more about the constructs than their measures, possibly only “Network acculturation…” (not “a measure of”); OR change all Hs to be about relationships between measures of constructs rather than the constructs themselves
- One of my main recommendations for the paper is to clarify the final argument and the support for the RQs. First, the fact that little research has been done on an area is not as strong of a reason for doing it as noting why such research should be done. Second, I don’t get a strong sense in the argument about why we might expect Network acculturation to be negatively related to Latinx enculturation (etc.) (H1), why you expect an indirect relationship in H3, and why acculturation and enculturation will be related to help-seeking as you suggest in H4 (evidence for H2 seems clear). One possibility is to tighten up intro before the first heading (keeping deeper introduction of terms for the sections of the review of lit) to make room for a summary of the argument here that clearly leads to each RQ. It looks like the last phrase of H4 also appears in h2. Prob make H4 just about help-seeking.
METHOD: The overall discussion sounds quite reasonable.
- 238: Explain what you mean by “secondary analysis.” Probably explain “egocentric” data, just in case the reader is not familiar with network theory terms. There is a good rationale for the Prolific® data collection.
- 251: “which guided the present study”—how did removing participants who timed out “guide” this study? (maybe just delete the phrase)
- 257: Larger sample that aligns with [add “s”]. What do you mean by “that aligns with general estimates”?
- 259: “Followed evidence-based practices.” Explain precisely what you mean (as you do below). Personally, I don’t like the phrase “evidence-based practices” as it seems just a buzzword meant to give the research credibility in the eyes of some readers. Your study will already have credibility—and more clarity—if you just mention the practices rather than label them as “evidence-based.” But that is your choice.
- 261: The study uses “social exchange theory”? How? This was not mentioned anywhere above. Maybe delete and just say “standard practices regarding recruitment methods…”
- 269ff: Clear on construction of measure, and the measure has a high reliability.
- 288: If possible, find a way to introduce name generators and name interpreters when you first mention them, rather then using them in text with definitions a few lines below.
- 306 ff: Clear on measures of enculturation and acculturation (here, at last, how you are using these terms becomes clear).
- 346: If item was negatively related, why not just reverse code it?
- 363ff: I will be honest to say that these statistics are not my expertise. Please rely on other reviewer for comments here. For example, is it common to use all five fit indices? (373-375)
- Just a thought: The definitions of social capital and network adjustment both seem to involve the number of people/connections one knows to give them support. Are these terms overlapping? When you mention the second term above, note how it is different from the first.
RESULTS
- 402-403: “…were examined descriptively by generational status, a proxy for acculturation”: Clarify what you mean by “generational status”
- Bivariate analysis findings are clearly stated.
- 437: Probably add “White American” before “acculturation,” for parallelism and clarity of meaning.
- Results are clear overall.
DISCUSSION
- 481: “Many hypotheses were supported.” I saw only 4 hypotheses listed.
- 489-490: Wha do the findings here mean practically. The language is clear, but the application could be clearer. Does this suggest, for example, that social capital might be highest for those Latinx individuals who have friend groups in both Latinx and White American communities? You do this well at lines 522 ff.
- 494-495: Same point: Give a plain English implication for the finding.
- 502: The idea of “pressure to acculturate” is also in Young Yun Kim’s (2001 book, 2005 and several other places) where she includes “pressure to conform” in her theory of acculturation.
- 513: Prob add “social” before “class,” just for clarity.
- 520: The rest of the discussion is excellent—clear summaries of findings with clear implications.
Author Response
We want to thank the reviewers for their comments. We hope that they find that we have addressed their concern or provided a suitable response that answers their questions. Ultimately, we believe that their contribution has strengthened the quality of the manuscript. We look forward to your response.
Solid abstract—quite clear
- Thank you.
Introduction introduces key concepts clearly, with good definitions and clear documentation in previous research.
- Again, we thank the reviewer for their feedback.
88: I was a little unsure what you mean by “the cascading of effects…early in life”. The use
of “the” suggests you’ve introduced or talked about such a cascade before.
- We changed the wording from cascading to compounding. We think this provides a better description of what we were trying to convey.
91-92: “acculturation…assimilation.” The terms are used as if they have already been defined. Without prior context of some sort, the reader may not know how you are using these seemingly similar terms (i.e., are these different things?). I had a question about the next sentence (“However…” and how it related, but definitions of the terms, as I think you are using them, would make the “however” sentence clear.
- We did provide an initial definition and highlighted it. However, we also tried to expand it. As we are sure the reviewer is aware, the acculturation literature can be pretty dense, with layers of definitions that are difficult to unpack. For example, even in this initial definition, we referenced assimilation as a strategy. This references a specific and heavily used theory of acculturation, which further specifies how these strategies translate to specific acculturative experiences. It’s difficult to unpack each aspect of that without launching into a full discussion of acculturation which comes later. We did our best to provide additional content that further supports this discussion.
- (p. 2, lines 53-68) Acculturation is defined as the cultural changes that occur when two distinct cultural groups interact for a sustained period, resulting in changes in one or both groups (Redfield et al., 1936). Acculturation has been described as degrees of change associated with strategies (e.g., assimilation, integration, separation, and marginalization) enacted by individuals and their communities that create specific acculturative experiences (Berry & Sam, 2016).
102: “despite their diverse origins and applications.” I’m not sure what you mean. Are you
beginning a list of description of different types of adjustment with “psychological
acculturation”? If so, preview the list so the reader knows where the argument is going.
- We agree with the reviewer that the language was awkward and suggestive of something else. We removed the language and added to the sentence to improve the flow.
- (p. 3, lines 104-105) Cross-cultural psychological theories have been relatively dominant in describing acculturation experiences.
Is this distinction your own (as it seems from the writing), or is this based on Berry’s well
known distinctions? If the list of types of acculturation is your own, use language to suggest
that (e.g., various sources suggest three main types of acculturation relevant to this study,”
or something like that).
- Thank you for pointing this out. This was an oversight. It was intended to continue Berry’s work. We added a different but equally supportive citation. This section has also been substantially rewritten to incorporate other suggestions identified by the reviewers.
114: “Social relationships are ore to how acculturation is described and understood.” Provide evidence for this claim. I know that years ago, some scholars in my own field
(communication) said that ability to form close relationships was one of the main aspects of “intercultural communication competence” (Hammer, Gudykunst, & Wiseman, 1978), but you probably have your own sources. The current evidence (Rodriguez et al., 2002, etc.) is about measures that scholars have introduced, which doesn’t establish the connection between social relationships and acculturation that findings would
- We included better citations supporting the connection between acculturation and social relationships.
- (p. 3, lines 138-141) Social relationships are a critical part of the interaction facilitating acculturation (Berry, 2003; Berry & Sam, 2016; Masgoret & Ward, 2006; Ward & Geeraert, 2017) and may serve as an important determinant of acculturative experiences.
122-123: “with whom a person shares or diverts from culturally”: I’m not sure what you mean by “diverts” here, so maybe another word. And because of the structure of the sentence, the section action has two prepositions (with whom diverts from)à “…social relationships in which a person approaches or moves away from others culturally…”?
- We agreed that the wording was odd. We have made adjustments to the sentence to add clarity.
- (p. 4, lines 158-160) Network acculturation refers to the interconnected social relationships that form a part of a person’s social network and contribute to their acculturation experiences (Vacca et al., 2018).
127: “For example…strategies.” Again, strategies are mention that have not been previously outlined, so they lack context.
- We have added content in the introduction and the literature review section that more clearly defines the meaning of the strategies.
- (p. 2, lines 53-59) Acculturation has been described as degrees of change associated with strategies (e.g., assimilation, integration, separation, and marginalization) enacted by individuals and their communities that create specific acculturative experiences (Berry & Sam, 2016).
- (p. 3, lines 109-123) . According to Berry (2003), individuals develop strategies to preserve a desired amount of their heritage culture while engaging in a particular level of interaction with members of the cultural majority. High and low retention levels and interaction topologically form four strategies (e.g., assimilation, integration, separation, or marginalization) that reflect a person's social integration and cultural change (Berry & Sam, 2016). For example, with assimilation, individuals desire to engage in high levels of interaction with individuals from the cultural majority while minimizing one's heritage (Berry, 2006). Conversely, individuals who desire to maintain their cultural heritage while minimizing interaction with the cultural majority use separation as their acculturation strategy (Berry, 2006), reflecting their initial socialization to their heritage culture (i.e., enculturation; Weinreich, 2009). Integration occurs when individuals seek a balance or high levels of cultural retention and interaction with individuals from the cultural majority (Berry, 2003). Lastly, Berry (2006) suggests that individuals who minimize the retention of their cultural heritage but are rebuffed by individuals from the cultural majority are marginalized.
- For many groups, integration has been identified as the most adaptive acculturation strategy (Choy et al., 2021), but not for Latinos/as. Historically, separation has often been the most adaptative strategy because of the Latino/a diaspora (i.e., resettlement in traditional locations with dense Latino/a populations) in the U.S. and protective factors (e.g., social support) associated with Latino/a culture (Ferguson & Birman, 2016). More recently, Latinos/as have resettled in U.S. communities with previously small or absent Latino/populations (Kochnar et al., 2005), which is associated with a greater need for adaptation and integration (Masked for Review, 2024). Therefore, other acculturation may be more viable as means of adapting than before, requiring continued investigation into the most adaptive strategies for Latinos/as and under what circumstances.
Note: There is a lot of self-citation here.
- We have reworked this section to reduce some of the self-citation. We have completed a significant amount of work in this area, which we sought to elaborate on in this and subsequent studies, so some citation remain.
157-158: “social capital . . . identifies the pathways…”: unusual wording
- We changed the wording.
- (p. 5, lines196-198) Social capital is recognized as a key determinant of QoL (Nutakor et al., 2023), suggesting that a person's social environment offers benefits across areas of life that are important to perceived quality and satisfaction (Buijs et al., 2016).
160: “social capital has demonstrated inconsistent results”—in terms of what?
- We changed the wording.
- (p. 5, lines 198-200) Despite its salutogenic effects, social capital has demonstrated an inconsistent relationship to QoL, often relating to a person’s mental health but not physical health (Gao et al., 2018).
166-167: “increased exposure to American culture changes [add ‘s’] important elements..— which elements?
- Here, we are referring to the size and frequency of interaction described in the prior part of the sentence. We have deleted reference to elements, which made it seem like we were inferring something different.
- (p. 5, lines 203-206) For example, Keefe (1984) reported that the familial network size and frequency of interaction of Mexican Americans generationally decreased, suggesting that increased exposure to American culture changes the qualities of a person's social networks.
189: Does “it” here refer to culture or to Arnault’s model?
- We add the model name to improve the clarity.
- (p. 5, lines 228—231) Additionally, Saint Arnault's model identifies the importance of a person's social network as a driving mechanism for help-seeking, suggesting that the formulation of social capital (i.e., network size, trust, and knowledge of help-seeking pathways) occurs within a specific cultural context through a person's social relationships.
194, 200: “enculturation…separation…acculturation…” Terms need definition or introduction, probably above in the paper.
- We have provided definitions for these terms in our revisions.
- (p. 3, lines 109—123) According to Berry (2003), individuals develop strategies to preserve a desired amount of their heritage culture while engaging in a particular level of interaction with members of the cultural majority. High and low retention levels and interaction topologically form four strategies (e.g., assimilation, integration, separation, or marginalization) that reflect a person's social integration and cultural change (Berry & Sam, 2016). For example, with assimilation, individuals desire to engage in high levels of interaction with individuals from the cultural majority while minimizing one's heritage (Berry, 2006). Conversely, individuals who desire to maintain their cultural heritage while minimizing interaction with the cultural majority use separation as their acculturation strategy (Berry, 2006), reflecting their initial socialization to their heritage culture (i.e., enculturation; Weinreich, 2009). Integration occurs when individuals seek a balance or high levels of cultural retention and interaction with individuals from the cultural majority (Berry, 2003). Lastly, Berry (2006) suggests that individuals who minimize the retention of their cultural heritage but are rebuffed by individuals from the cultural majority are marginalized.
225: Probably add “A” before “network measure” (H1). But since later Hs are more about the constructs than their measures, possibly only “Network acculturation…” (not “a measure of”); OR change all Hs to be about relationships between measures of constructs rather than the constructs themselves
- Thank you for pointing this out. We have removed the reference to the measure and framed the hypothesis in terms of the construct.
- (p. 6, lines 271-272) H1: Network acculturation will be negatively related to Latino/a enculturation and positively related to White American acculturation.
One of my main recommendations for the paper is to clarify the final argument and the support for the RQs. First, the fact that little research has been done on an area is not as strong of a reason for doing it as noting why such research should be done. Second, I don’t get a strong sense in the argument about why we might expect Network acculturation to be negatively related to Latinx enculturation (etc.) (H1),
- We are building on our previous work in other studies, which has found this association. We add this to the manuscript below to establish a clear connection with H1.
- (p. 4, 164--170) Language use across stressful relationships has been used to measure network acculturation, which demonstrated a negative association with psychological measures of Latinx enculturation (i.e., socialization to one's heritage culture) and a positive association with White American acculturation, suggesting that network acculturation may serve as a potential determinant for psychological experiences (Masked for Review, 2021; Masked for Review, 2024), highlighting the importance of social relationships in determining acculturation experiences.
why you expect an indirect relationship in H3,
- We have continued to build on our previous work in this area. One of the shortcomings of previous acculturation research was the assumption that acculturation, by itself, was associated with health and mental health outcomes. Researchers have since incorporated stress process models that further delineate these relationships. Similarly, we have argued and continue to investigate that acculturative processes contribute to social capital and provide access to resources that enhance other aspects of life. We have included the following to provide additional support.
- (pp. 4-5, lines 185—198) Social relationships are an important bridge between acculturation, social capital, and social networks (Allen et al., 2014; Concha et al., 2014; Keefe, 1984; Rodionov, 2021; Valencia-Garcia et al., 2012). Acculturation has been identified as a factor that changes a person's relationships and social networks (Keefe, 1984). Valencia-Garcia and colleagues (2012) found that higher levels of acculturation were associated with greater social capital and reported access to needed services. Park and colleagues (2023) reported a more conditional relationship with acculturation moderated by social cohesion (i.e., a proxy for social capital) and negative social interaction, with higher acculturation and social cohesion associated with better mental health and higher negative interaction and acculturation associated with worse mental health. In the context of acculturation, integration into both cultural groups may provide greater access to resources that maximize a person’s social capital. Identifying the role that acculturation plays in altering social capital is critical to understanding how acculturative experiences contribute directly and indirectly to a person's quality of life.
and why acculturation and enculturation will be related to help-seeking as you suggest in H4 (evidence for H2 seems clear).
- Acculturation has been connected to help-seeking in the literature which has suggested difference in formal and informal use by acculturation level. Rather than suggesting one or the other, we are suggesting that access to informal and formal channels of assistance may enhance a person’s willingness and ability to seek assistance. Additionally, the help seeking measure is composed of formal and informal sources, so it would seem to suggest that those who are integrated may successfully navigate both.
- (pp. 4-5, line 185-198) Social relationships are an important bridge between acculturation, social capital, and social networks (Allen et al., 2014; Concha et al., 2014; Keefe, 1984; Rodionov, 2021; Valencia-Garcia et al., 2012). Acculturation has been identified as a factor that changes a person's relationships and social networks (Keefe, 1984). Valencia-Garcia and colleagues (2012) found that higher levels of acculturation were associated with greater social capital and reported access to needed services. Park and colleagues (2023) reported a more conditional relationship with acculturation moderated by social cohesion (i.e., a proxy for social capital) and negative social interaction, with higher acculturation and social cohesion associated with better mental health and higher negative interaction and acculturation associated with worse mental health. In the context of acculturation, integration into both cultural groups may provide greater access to resources that maximize a person’s social capital. Identifying the role that acculturation plays in altering social capital is critical to understanding how acculturative experiences contribute directly and indirectly to a person's quality of life.
One possibility is to tighten up intro before the first heading (keeping deeper introduction of terms for the sections of the review of lit) to make room for a summary of the argument here that clearly leads to m.each RQ. It looks like the last phrase of H4 also appears in h2. Prob make H4 just about help-seeking.
- We have made the suggested change to the hypothesis.
- H4: White American acculturation will be negatively while Latino/a enculturation and will be positively related to help-seeking.
METHOD: The overall discussion sounds quite reasonable.
238: Explain what you mean by “secondary analysis.” Probably explain “egocentric” data,
just in case the reader is not familiar with network theory terms. There is a good rationale for the Prolific® data collection.
- (p. 7, lines 291—296) The original survey design focused on the acculturation, negative social exchange networks, and mental health of Latino/a adults. However, a secondary analysis (e.g., examination of existing data to test a different set of hypotheses than initially intended) was conducted using cross-sectional egocentric social network data. Egocentric approaches ask a person (i.e., identified as ego) to list several people in their social network to describe potential network relationships.
- 251: “which guided the present study”—how did removing participants who timed out
“guide” this study? (maybe just delete the phrase)
- Statement was removed.
257: Larger sample that aligns with [add “s”]. What do you mean by “that aligns with general estimates”?
- We made some changes to the sentence to add some clarity.
- (p. 8, lines 315-317) However, sample sizes between 200 and 300 are considered minimally sufficient to detect meaningful effects in structural equation models (Harrington, 2009; Kline, 2005). Therefore, we recruited a sample of that size.
259: “Followed evidence-based practices.” Explain precisely what you mean (as you do
below). Personally, I don’t like the phrase “evidence-based practices” as it seems just a
buzzword meant to give the research credibility in the eyes of some readers. Your study will
already have credibility—and more clarity—if you just mention the practices rather than
label them as “evidence-based.” But that is your choice.
- Here we are referencing the practices outline by Dillman. Dillman has spent substantial time testing different aspects of survey designs to identify those practices that compromised or enhanced recruitment and retention and minimize errors. We tried to clarify and added some content.
- (p. 8, lines 318-323) The construction of the survey and its implementation followed evidence-based practices outlined by the Tailored Design Method (Dillman et al., 2014). This approach uses standardized recruitment methods and evidence-based survey practices (e.g., the content of the recruitment message, the cognitive flow of the survey content, and question types and placement) to minimize survey errors (e.g., nonresponse) and maximize recruitment and retention (Dillman et al., 2014).
261: The study uses “social exchange theory”? How? This was not mentioned anywhere
above. Maybe delete and just say “standard practices regarding recruitment methods…”
- (We took the reviewer’s suggestion and made the change.
269ff: Clear on construction of measure, and the measure has a high reliability.
288: If possible, find a way to introduce name generators and name interpreters when you first, mention them, rather than use them in text with definitions a few lines below.
- We searched the document and found that the first reference to name generators and interpreters was the reference identified by the reviewer. I’m not sure if the reviewer was asking us to reorganize within the paragraph or to introduce name generators earlier in the paper. Consistent with the reviewer's line reference, we made the adjustment within the paragraph but can make other adjustments if necessary.
- (p. 9, 359—368) Name generators emphasize a person's network's structure (e.g., size), followed by additional questions (e.g., name interpreters) that provide information about the network's composition. Using name generators and interpreters, network measures focusing on language use have been used to measure network aspects of acculturation and a potential determinant of psychological acculturation (Masked for Review, 2021; Masked for Review, 2024). Therefore, the current study used name generators and name interpreters to assess different aspects of respondents' social networks relative to acculturation. The current study placed greater emphasis on the composition of respondents' networks, consistent with other network studies examining acculturation. These aspects have demonstrated a unique operationalization that is distinct from traditional psychological measures (see Masked for Review, 2024).
306 ff: Clear on measures of enculturation and acculturation (here, at last, how you are using these terms becomes clear).
- (p. 4, lines 104-123). We did try to add content earlier to introduce these concepts earlier than the measurement section.
346: If item was negatively related, why not just reverse code it?
- It’s a good point. Initially, we looked for more guidance on scoring but did not find any clear direction. Given the negative finding, we followed the same procedures as Wilson et al., 2005. Also, reverse coding this particular item does not necessary translate to an increased likelihood to seek help. The item was included to capture complete help-negation rather than intention to seek help, which is why it likely relates negatively to other items and, therefore, represents something conceptually related to but different.
363ff: I will be honest to say that these statistics are not my expertise. Please rely on other
reviewer for comments here. For example, is it common to use all five fit indices? (373-375)
- We have reported the typical indices discussed in the SEM literature. More recent publications have only reported on a couple of indices. However, we have presented all appropriate indices to allow readers to make the best judgment.
Just a thought: The definitions of social capital and network adjustment both seem to involve the number of people/connections one knows to give them support. Are these terms overlapping? When you mention the second term above, note how it is different from the first.
- Yes, there is an overlap between the two concepts, which is one reason that we have linked them together. While race and ethnicity are related to social capital, the social capital and acculturation literature have not sufficiently tied these two together. To create distinctions, we opted to use the compositional measure rather than structural measures.
RESULTS
402-403: “…were examined descriptively by generational status, a proxy for acculturation”:
Clarify what you mean by “generational status”
- We added to the measures section to provide a description of generational status and noted in the analytic plan.
- (p. 8, lines 326-338) Generational status was once used as a proxy for acculturation and is still helpful in demographically understanding where people are situated in their family’s migration history (Phinney, 2003; Thomson & Hoffman-Goetz (2017). Participants were asked which of the following best described their generational status, with response options ranging from 1st to 4th The following definitions were provided to participants. First-generation is defined as those participants who report nativity outside of the U.S. Those reporting second-generation report nativity in the U.S. but have parents who report nativity outside of the U.S. Third-generation participants are those who are reporting nativity U.S. for themselves and their parents but report nativity outside of the U.S. for their grandparents. Lastly, fourth-generation participants report nativity in the U.S. for themselves, their parents, and their grandparents.
Bivariate analysis findings are clearly stated.
- Thank you.
437: Probably add “White American” before “acculturation,” for parallelism and clarity of meaning.
- We separated the headings in this section to better identify the network measure of acculturation from the psychological measures. Additionally, we added a heading further delineating the enculturation and White American Acculturation to create parallelism and contribute to the manuscript's clarity.
- (p. 9, lines 357, 381)
Results are clear overall.
DISCUSSION
481: “Many hypotheses were supported.” I saw only 4 hypotheses listed.
- The sentence was poorly constructed and was removed.
489-490: What do the findings here mean practically? The language is clear, but the application
could be clearer. Does this suggest, for example, that social capital might be highest for those
Latinx individuals who have friend groups in both Latinx and White American communities?
You do this well at lines 522 ff. • 494-495: Same point: Give a plain English implication for the finding. We added a plain
- We have made several edits to the discussion to include some practical discussion of the finds.
513: Prob add “social” before “class,” just for clarity.
- Change made. Thank you.
520: The rest of the discussion is excellent—clear summaries of findings with clear implications.